# Dual-stream Network for Visual Recognition

**Mingyuan Mao[1,†], Peng Gao[3,†], Renrui Zhang[2,†], Honghui Zheng[3], Teli Ma[2],**
**Yan Peng[3], Errui Ding[3], Baochang Zhang[1,*], Shumin Han[3,*]**
[1]Beihang University, Beijing, China
[2]Shanghai AI Laboratory, China
[3]Department of Computer Vision Technology (VIS), Baidu Inc
[*]Corresponding author, email: bczhang@buaa.edu.cn, hanshumin@baidu.com
[†]Equal contributions

## Abstract

Transformers with remarkable global representation capacities achieve competitive results for visual tasks, but fail to consider high-level local pattern information in input images. In this paper, we present a generic Dual-stream Network (DS-Net) to fully explore the representation capacity of local and global pattern features for image classification. Our DS-Net can simultaneously calculate fine-grained and integrated features and efficiently fuse them. Specifically, we propose an Intra-scale Propagation module to process two different resolutions in each block and an Inter-Scale Alignment module to perform information interaction across features at dual scales. Besides, we also design a Dual-stream FPN (DS-FPN) to further enhance contextual information for downstream dense predictions. Without bells and whistles, the proposed DS-Net outperforms DeiT-Small by 2.4% in terms of top-1 accuracy on ImageNet-1k and achieves state-of-the-art performance over other Vision Transformers and ResNets. For object detection and instance segmentation, DS-Net-Small respectively outperforms ResNet-50 by 6.4% and 5.5 % in terms of mAP on MSCOCO 2017, and surpasses the previous state-of-the-art scheme, which significantly demonstrates its potential to be a general backbone in vision tasks. The code will be released soon.

## 1  Introduction

In recent years, convolutional neural networks (CNNs) have dominated various vision tasks including but not limited to image recognition [22, 18, 38, 39, 5, 19, 41, 20, 52], object detection [37, 2, 26, 36, 24, 51, 30, 21, 11] and segmentation [17, 28], thanks to their unprecedented representation capacity. However, limited receptive fields of convolutions inevitably neglect the global patterns in images, which might be crucial during inference. For example, it is more likely a chair than a elephant nearby a table. Such object-level information cannot be fully explored by local convolutions, that hampers further improvement of CNNs. Motivated by the success of Transformer architecture [43] in Natural Language Processing (NLP) [9, 34, 1] and Multi-modality Fusion [29, 40, 12], researchers are trying to apply Transformer to vision tasks and have obtained promising results. In [10, 55, 48, 4, 7, 45, 46, 27, 3, 54, 14, 15, 44, 53], many novel architectures and optimizing strategies are proposed and achieve comparable or even better performance than CNNs, emphasizing the significance of extracting global features in vision tasks.

Inspired by the verified efficacy of CNN and Transformer, many concurrent works, such as Cont-Net [47] a CvT [46], attempt to introduce convolutions to vision transformers in different manners, hoping to combine both advantages. In CvT [46], linear projection of self-attention block in the Transformer module are replaced with convolutional projection. In contrast, ContNet[47] performs

convolution on different token maps to build mutual connections, which is similar to Swin Transformer [27].

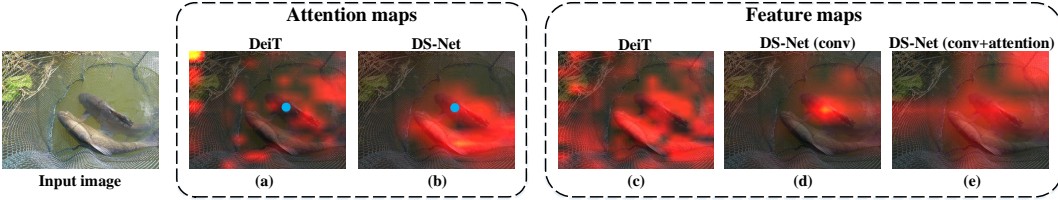

Figure 1: Comparison of attention maps and feature maps of DeiT and our proposed DS-Net, both of which are obtained from the last block. (a) and (b) are attention weights of the blue points in the images. To achieve clearer illustration, the figures are acquired by overlaying heatmaps to the input images.

However, some drawbacks of existing models still remain to be solved. Firstly, works mentioned above either perform convolution and attention mechanisms sequentially, or just replace linear projection with convolutional projection in attention mechanisms, which may not be the most ideal design. Additionally, the conflicting properties of such two operations, convolution for local patterns but attention for global patterns, might cause ambiguity during training, which prevents their merits from merging to the maximum extent. Furthermore, self-attention can capture long-range information via the built-in all-pair interaction in theory, but it is quite possible that attentions might be confused and disturbed by neighboring details in high resolution feature maps, fail to build up object-level global patterns(see Fig. 1(a)). Finally, the computation cost of self-attention is unaffordable due to the quadratic computation complexity of sequence length. Although PVT [45] and APNB [56] downsample the key-query features to improve the efficiency of self-attention operator, they both abandon fine-graind local details of the image, which greatly impairs their performance.

In this paper, we address the these issues by introducing a *Dual-stream Network (DS-Net)*. Instead of single stream architecture as previous works, our DS-Net adopts Dual-stream Blocks (DS-Blocks), which generates two feature maps with different resolutions, and retains both local and global information of the image via two parallel branches. We propose a Intra-scale Propagation module here to process two feature maps. Specifically, high-resolution features are used to extract fine-grained local patterns with depth-wise convolution, while low-resolution features are expected to summarize long-range global patterns. Considering low-resolution features themselves contain more integrated information, it would be much more easier for self-attention mechanism to capture object-level patterns rather than overwhelmed by trivial details (see Fig. 1(b)). Such dual-stream architecture disentangles local and global representations, which helps maximize both their merits, and thus generates better representations compared to DeiT baseline (see (c) and (e) in Fig. 1). Besides, low-resolution feature maps for self-attention dramatically reduce the memory cost and computation complexity. After parallelly processing dual streams, we present Inter-scale Alignment module based on co-attention mechanism at the end of DS-Blocks. This is because local details and global patterns capture different perspectives of the image, which are misaligned not only in pixel positions but also in semantics. Hence, this module is designed for modeling complex cross-scale relations and adaptively fuse local and global patterns.

Besides, we apply our DS-Blocks to Feature Pyramid Networks for further feature refinement, named DS-FPN. In this way, multi-level features are capable of extracting contextual information from both local and global views, improving performance of downstream tasks. This demonstrates that our Dual Stream design could be utilized as a plug-in building block not only for image recognition, but for many other vision tasks.

The contributions of this work are concluded as follows:

1. We present a novel Dual-Stream Network, named DS-Net, which retains both local and global features in DS-Block. The independent propagation maximize the advantages of convolution and self-attention, thus eliminating the conflicts during training.

2. We propose Intra-scale Propagation and Inter-Scale Alignment mechanism to achieve effective information flows within and between features of different resolutions, thus generating better representations of the image.

3. We introduce Dual-stream Feature Pyramid Network (DS-FPN) to enhance contextual information for downstream dense tasks and achieves better performance with little extra costs.

4. Without bells and whistles, the proposed DS-Net outperforms DeiT baseline by significant margins in terms of top-1 accuracy on ImageNet-1k and achieves state-of-the-art performance over other Vision Transformers and CNN-based networks on image classification and downstream tasks, including object detection and instance segmentation.

## 2   Related work

**Vision Transformers.** Motivated by the great success of Transformer in natural language processing, researchers are trying to apply Transformer architecture to Computer Vision tasks. Unlike mainstream CNN-based models, Transformer is capable of capturing long-distance visual relations by its self-attention module and provides the paradigm without image-specific inductive bias. ViT [10] views $16 \times 16$ image patches as token sequence and predicts classification via a unique class token, which shows promising results. Subsequently, many works, such as DeiT [42] and PVT [45] achieve further improvement on ViT, making it more efficient and applicable in downstream tasks. ResT [50] proposed a efficient transformer model. Besides, models based on Transformer also give leading performance in many other vision tasks such as object tracking and video analysis.

**Local and Global Features in Transformers.** Despite the Transformer's superiority upon extracting global representations, image-level self-attention is unable to capture fine-grained details. To tackle this issue, previous works assert to obtain the local information at the same time and then fuse features of the two scales. TNT [16] proposes intra-patch self-attention block to model local structures and aggregates them with global patch embeddings. Swin Transformer [27] extracts local features within each partitioned window and fuses them by self-attention in successive shifted windows. Similarly, shifted windows are replaced by convolutions in ContNet [47]. Following such design of local windows, Twins [6] uses inter-window attention to aggregate global features. Swin Transformer and Twins both give advanced performance, but still rely on local-window self-attention to capture fine-level features, which works intuitively worse than CNN. Besides, Swin's shifted windows are complex for device optimization and Twins [6] recurrently local and global attentions would constrain each other's representation learning.

**Equip Transformers with Convolution.** Compared to self-attention mechanism, CNN has its unique advantages of local modeling and translation invariance. Therefore, some existing transformers explore the hybrid architecture to incorporate both merits for better visual representation. T2T [48] progressively aggregate neighboring tokens to one token to capture local structure information, similar to convolution operation. Followingly, CvT [46] designs Convolutional Token Embedding and Convolutional Transformer Block for capturing more precise local spatial context. CPVT [7] utilizes one convolutional layer to dynamically generate positional encodings, adaptive for varying image scales. Conformer [33] combines transformer with an independent CNN network for specializing local features and flows information via lateral connections. Nonetheless, none of the aforementioned models maximizes the potential of the hybrid architecture. They either use CNN for marginal operations to boost Transformer, *e.g.*, tokens generation in CvT [46] and positional encodings in CPVT [7], or view CNN as a relatively separate branch, like Conformer.

Different from previous works, in proposed DS-Blocks, we treat self-attention and convolution as dual resolution processing paths via Intra-scale Propagation module, where the former aims to extract local fine-grained details, and the latter focus on exploring features from a global view. On top of that, their information is reasonably fused by Inter-scale Alignment module based on co-attention, which addresses the issue of feature misalignment. By this design, the local and global patterns could be simultaneously extracted and the capability of CNN is fully exerted in Transformer architecture.

## 3   Method

### 3.1   Dual-scale Representations

The overall pipeline of our proposed DS-Net is shown in Fig. 2. Motivated by stage-wise design in previous networks like ResNet [18], we set 4 stages in the architecture, whose down-sampling factors

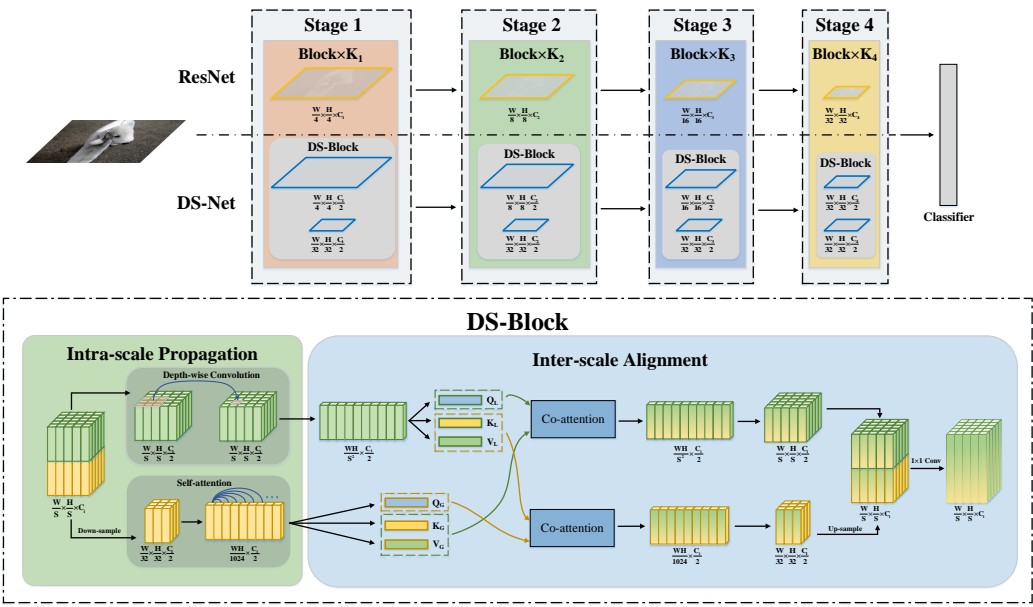

Figure 2: Illustration of the proposed DS-Net, including Intra-scale Propagation module and Inter-scale Alignment module. Compared to ResNet, in which only single resolution is processed, our DS-Net, instead, generates dual-stream representations via DS-Blocks.

are 4, 8, 16, 32, respectively. Within each stage, Dual-stream Blocks (DS-Blocks) are adopted to generate and combine dual-scale representations.

Our key idea is to keep local features in a relatively high resolution to reserve local details, while represent global features with a lower resolution (1/32 of the image size) to retain global patterns. Specifically, in each DS-block, we split the input feature map into two parts at the channel dimension. One is for extracting local features, denoted as $f_l$, the other is for summarizing global features, denoted as $f_g$. Notably, we keep the size of $f_g$ unchanged in all stages across the network by down-sampling it with a proper factor, $\frac{W}{32} \times \frac{H}{32} \times \frac{C_i}{2}$, where $W$ and $H$ represents the width and height of the input image, and $C_i$ represents the channel number of input features in current stage. Aided by the high resolution of $f_l$, the local patterns could be much reserved for subsequent extraction, but thanks to the low resolution of $f_g$, the exploration of non-local and object-level information is greatly benefited.

### 3.2 Intra-scale Propagation

Local and global features are representations of one image from two totally different views. The former focuses on fine-grained details, essential for tiny-object detection and pixel-level localization, while the latter aims at modeling object-level relations between long-range parts. Therefore, given dual-scale features $f_l$ and $f_g$, we perform Intra-scale Propagation module to parallelly process them.

**Local representation.** For high-resolution $f_l$ with the size of $W_i \times H_i \times C_l$, where $C_l$ equals $\frac{C_i}{2}$, we perform 3×3 depth-wise convolution to extract local features as follows and obtain $f_L$:

$$f_L(i,j) = \sum_{m,n}^{M,N} W(m,n) \odot f_l(i+m, j+n), \tag{1}$$

where $W(m,n), (m,n) \in (-1, 0, 1)$ represents the convolution filters, $W(m,n)$ and $f_l(i,j)$ are both $\frac{C_i}{2}$ dimensional vectors, $\odot$ denotes element-wise product. By the power of depth-wise convolution, $f_L$ is able to contain fine-grained local details of the input image.

**Global representation.** For low-resolution representation $f_g$ with the fixed size of $\frac{W}{32} \times \frac{H}{32} \times \frac{C_i}{2}$ as illustrated in Section 3.1, we first flatten $f_g$ to a sequence with the length of $l_g$, the product of $\frac{W}{32}$ and

$\frac{H}{32}$, in which each element is a $\frac{C_i}{2}$ dimensional vector. By doing this, each vector in the sequence is treated as a visual token without spatial information. Here, the dependencies between different token pairs are unrelated with their spatial positions in the feature map, which is totally different with the convolution. Then, we summarize the global information and model the object-level coupling relation via self-attention mechanism:

$$f_Q = f_g W_Q, \quad f_K = f_g W_K, \quad f_V = f_g W_V, \tag{2}$$

where $W_Q, W_K, W_V$ denote matrixes to generate queries, keys and values respectively. By calculating the similarity between $f_Q$ and $f_V$, we obtain attention weights for aggregating information from different locations of $f_g$. Finally, we calculate the weighted sum of attention weights and $f_V$, thus obtaining integrated features:

$$f_G = \mathrm{softmax}(\frac{f_Q f_K^T}{\sqrt{d}})f_V, \tag{3}$$

where $d$ equals $\frac{\frac{C_i}{2}}{N}$, $N$ denotes the number of attention head that we set to 1, 2, 5, 8 for 4 stages respectively in DS-Net.

Our dual-stream architecture disentangles fine-grained and integrated features in two pathways, which significantly eliminates the ambiguity during training. Additionally, the Intra-scale Propagation module processes feature maps with dual resolutions by two domain-specifically effective mechanisms separately, extracting local and global features to the maximum extent.

### 3.3 Inter-scale Alignment

A delicate fusion of dual-scale representations is vital for the success of DS-Net, since they capture two different perspectives of one image. To address this, a naive idea is to upsample the low-resolution representation using bilinear interpolation, and then fuse the dual-scale representations via 1 by 1 convolution after simply concatenating them position-wisely. Such naive fusion is not convincing enough. Furthermore, by visualizing the feature maps of dual-scale features, we observe that global features in low resolutions and local features in high-resolutions are actually misaligned (see (b) and (d) in Fig. 1). Therefore, considering the relation of two representations are not explicitly explored, it is unpersuasive to pre-define a fixed strategy to fuse them, such as concatenation, element-wise addition or production. Enlightened by [13, 32], we propose a novel co-attention-based Inter-scale Alignment module, whose scheme of Inter-scale Alignment module is shown in Fig. 2. This module aims to capture the mutual correlations between each local-global token pair, and propagate information bidirectionally in a learnable and dynamic manner. Such mechanism prompts local features to adaptively explore their relations with global information, enabling themselves to be more representative and informative, and vice versa.

Given extracted local features $f_L$ with the size of $W_i \times H_i \times \frac{C_i}{2}$ and global features $f_G$ with the size of $l_g \times \frac{C_i}{2}$, we first flatten $f_L$ to a sequence with the length of $l_l$, the product of $W_i$ and $H_i$, in which each element is a $\frac{C_i}{2}$ dimensional vector. $f_L$ and $f_G$ are now in the same format but with different length. Then we perform co-attention on two sequences as follows:

$$\begin{aligned} Q_L = f_L W_Q^l, \quad K_L = f_L W_K^l, \quad V_L = f_L W_V^l, \\ Q_G = f_G W_Q^g, \quad K_G = f_G W_K^g, \quad V_G = f_G W_V^g, \end{aligned} \tag{4}$$

where the sizes of $W$ are all $\frac{C_i}{2} \times dim$, and the $dim$ is a hyper-parameter. Thus we have the transformed features of local and global representations. Then we calculate the similarities between every pair of $f_L$ and $f_A$ to obtain the corresponding attention weights:

$$W_{G \to L} = \mathrm{softmax}(\frac{Q_L K_G^T}{\sqrt{d}}), \quad W_{L \to G} = \mathrm{softmax}(\frac{Q_G K_L^T}{\sqrt{d}}). \tag{5}$$

The size of $W_{G \to L}$ and $W_{L \to G}$ are $l_l \times l_g$ and $l_g \times l_l$ respectively. The non-linear function softmax is performed at the last dimension. $W_{G \to L}$ reflects the importance of different tokens in global features to the local tokens. Likewise, global features can also extract useful information from local features via $W_{L \to G}$. Instead of fixed fusion strategy that might bring constraints, what and how the

information transfers are automatically determined by features themselves here. We can then obtain hybrid features as:

$$h_L = W_{G \to L} V_G, \quad h_G = W_{L \to G} V_L, \tag{6}$$

where the size of $h_L$ and $h_G$ are $l_l \times dim$ and $l_g \times dim$ respectively. Then we add a $1 \times 1$ convolution layer after hybrid features to further fuse the channels and reshape them to $W_i \times H_i \times \frac{C_i}{2}$ and $\frac{W}{32} \times \frac{H}{32} \times \frac{Ci}{2}$.

Such a bidirectional information flow is able to identify cross-scale relations between local and global tokens, by which dual-scale features are highly aligned and coupled with each other. After this, we could safely upsample low-resolution representation $h_G$, concatenate it with high-resolution $h_L$ and perform 1 by 1 convolution for channel-wise dual-scale information fusion. At the end of the last block in stage 4, we add a fully-connected layer as the classifier to conduct classification.

### 3.4 Dual-stream Feature Pyramid Networks

Introducing contextual information into Feature Pyramid Networks(FPN) [23] has been explored by [49, 31]. However, previous methods often cause large extra memory and computation costs, due to their complicated architectures and utilized high resolution feature maps. Besides, Non-local contexts would miss local details, which is disastrous for tiny object detection and segmentation. Here, we apply our Dual-stream design into FPN, named Dual-stream Feature Pyramid Networks(DS-FPN), by simply adding DS-Blocks to every feature pyramid scale. In this way, DS-FPN is able to better attain non-local patterns and local details with marginal increased costs at all scales, which further enhance the performance of subsequent object detection and segmentation heads. This shows our DS-Net can not only serve as a backbone but also a general plug-in building block in many other vision architectures.

Similar to FPN [23], we take image features from various scales from the backbone as input, and output corresponding refined feature maps of fixed channels number by a top-down aggregation methods. Our structure is composed of bottom-up pathways, Dual-stream lateral connections, and top-down pathways. The bottom-up and top-down pathways follow the design of FPN, but lateral connection here adopts DS-Block to process features in dual scales via Intra-scale Propagation and Inter-scale Alignment. See Fig. 3.

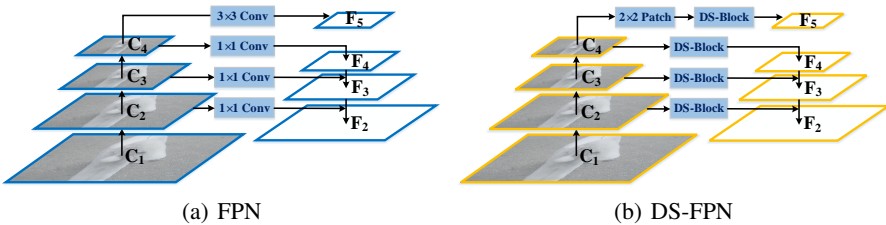

(a) FPN                                    (b) DS-FPN

Figure 3: The architecture of DS-FPN. $C_i$ denotes the feature maps in stages from backbone, and $F_i$ denotes the reconstructed features for detection and segmentation.

## 4 Experiments

In this section, we first provide three ablation studies to explore the optimal structure of DS-Net and interpret the necessity of dual-stream design. Then we give the experimental results of image classification and downstream tasks including object detection and instance segmentation. Specifically, We use ImageNet-1K [8] for classification and MSCOCO 2017 [25] for object detection and instance segmentation. All experiments are conducted on 8 V100 GPUs and the throughput is tested on 1 V100 GPU. As the number of blocks we set in different stages in the architecture (see Fig. 2) is flexible, we conduct experiments on 3 models with different parameter size, denoted as DS-Net-T (Tiny), DS-Net-S (Small) and DS-Net-B (Big), which have comparable parameters with ResNet-18, ResNet-50 and ResNet-101 respectively. The detailed setup is shown in Table 1.

Table 1: Detailed settings of DS-Net. Dconv denotes $3 \times 3$ depth-wise convolution, and MHSA denotes multi-head self-attention. $C_i$ denotes the number of channels in $ith$ stage. The feature dimension expansion ratio of each block is set to 4.

| Stage | Input size | DS-Net-T | | DS-Net-S | | DS-Net-B | |
|---|---|---|---|---|---|---|---|
| Stage 0 | 224×224 | 4×4, 64, stride=4, padding=0 | | | | | |
| Stage 1 | 56×56 | Dconv
MHSA − 8
$C_1 = 64$ | × 2 | Dconv
MHSA − 8
$C_1 = 64$ | × 3 | Dconv
MHSA − 8
$C_1 = 64$ | × 3 |
| Stage 2 | 28×28 | Dconv
MHSA − 8
$C_2 = 128$ | × 2 | Dconv
MHSA − 8
$C_2 = 128$ | × 4 | Dconv
MHSA − 8
$C_2 = 128$ | × 4 |
| Stage 3 | 14×14 | Dconv
MHSA − 8
$C_3 = 320$ | × 4 | Dconv
MHSA − 8
$C_3 = 320$ | × 8 | Dconv
MHSA − 8
$C_3 = 320$ | × 28 |
| Stage 4 | 7×7 | Dconv
MHSA − 8
$C_4 = 512$ | × 1 | Dconv
MHSA − 8
$C_4 = 512$ | × 3 | Dconv
MHSA − 8
$C_4 = 512$ | × 3 |
| | 7×7 | global average pooling, 1000-d fc, softmax | | | | | |

Table 2: DS-Net-T performance on ImageNet-1k validation set with different $\alpha$.

| $\alpha$ | 0 | 0.25 | 0.5 | 0.75 | 1 |
|---|---|---|---|---|---|
| Top-1(%) | 77.1 | 78.0 | **78.1** | 77.9 | 77.6 |
| Top-5(%) | 93.3 | **94.1** | **94.1** | 94.0 | 93.9 |
| Params (M) | 8.6 | 8.7 | 9.1 | 9.8 | 10.7 |
| FLOPs (G) | 1.573 | 1.578 | 1.592 | 1.615 | 1.647 |
| Throughput (Images/s) | 3240 | 1733 | 1199 | 912 | 740 |

## 4.1 Ablation Study

### 4.1.1 Ratio of Local to Global Features

**Settings.** As described in Section 3.1, we split the features into two parts in channel dimension as local and global representations, which are later processed by convolution and self-attention, respectively. We conduct an experiment to explore a appropriate partition ratio. We use $\alpha$ to denote the proportion of channel number of $f_g$ (global features) to the total channel number. When $\alpha$ equals 0, only depth-wise convolution is performed, and when $\alpha$ equals 1, only self-attention is performed. Noting that in this experiment, we choose DS-Net-T as the testing model, which simply concatenates $f_L$ and $f_A$ at the end of each block, without Inter-scale Alignment module.

**Results.** The results in Table 2 indicates that when $\alpha$ equals 0.5, DS-Net-T achieves the highest accuracy on ImageNet-1k validation set, which, to some extent, implies that local and global information play equally important roles for visual representations. Thus, in the following experiments, we equally split features into two parts at the channel dimension to obtain dual-scale representations.

### 4.1.2 None is Dispensable in DS-Block

**Settings.** Here, we illustrate that every component within DS-Block plays a vital role in the prediction, including either pathway of Intra-scale Propagation and bidirectional co-attention of Inter-scale Alignment. In Table 3, we respectively remove one of the four modules in every block of DS-Net-T* (DS-Net* represents the corresponding DS-Net version with Inter-scale Alignment module), while remain the others the same. Therein, $w/o\ f_L$ denotes that $f_l$ split at the beginning is directly fed to Inter-scale Alignment Module without exploring local features, and $w/o\ G \rightarrow L$ denotes that co-attention from global features to local features is not implemented, analogous to $w/o\ f_G$ and $w/o\ L \rightarrow G$. The version $w/o\ L \leftrightarrow G$ removes the entire co-attention, which equals DS-Net-T.

Table 3: Ablations of removing components of DS-Net-T$^*$ on ImageNet-1k validation set.

| Versions | DS-Net-T$^*$ | $w/o\ f_L$ | $w/o\ f_G$ | $w/o$ $G \to L$ | $w/o$ $L \to G$ | $w/o$ $L \leftrightarrow G$ |
|---|---|---|---|---|---|---|
| Top-1(%) | **79.0** | 76.7 | 76.6 | 76.5 | 76.4 | 78.1 |
| Top-5(%) | **94.8** | 93.6 | 93.7 | 93.5 | 93.4 | 94.1 |

Table 4: Comparison with the accuracy of other state-of-art methods on ImageNet-1k validation set. The input images are reshape to $224 \times 224$ resolution. DS-Net$^*$ represents the corresponding DS-Net version with Inter-scale Alignment module.

| Method | Params (M) | FLOPs (G) | Throughput (Images/s) | Top-1 (%) |
|---|---|---|---|---|
| ConvNet | | | | |
| ResNet-18 [18] | 11.8 | 2 | - | 69.9 |
| ResNet-50 [18] | 25.6 | 4.1 | - | 74.2 |
| ResNet-101 [18] | 44.5 | 7.8 | - | 77.4 |
| RegNetY-8GF [35] | 39.2 | 8 | - | 79.9 |
| RegNetY-16GF [35] | 83.6 | 15.9 | - | 80.4 |
| Transformer / Hybrid | | | | |
| DeiT-T [42] | 6 | - | 2536 | 72.2 |
| CPVT-Ti [7] | 6 | - | - | 72.4 |
| T2T-ViT-12 [48] | 6.9 | - | - | 76.5 |
| ConTNet-S [47] | 10.1 | 1.5 | - | 76.5 |
| DS-Net-T (**ours**) | 9.1 | 1.6 | 1199 | **78.1** |
| DS-Net-T$^*$ (**ours**) | 10.5 | 1.8 | 1034 | **79.0** (+6.8) |
| DeiT-S [42] | 22.1 | 4.6 | 940 | 79.9 |
| CrossViT-15 [4] | 27.4 | 5.8 | 640 | 81.5 |
| T2T-ViT-14 [48] | 22 | 5.2 | - | 81.5 |
| ConTNet-M [47] | 19.2 | 3.1 | - | 80.2 |
| TNT-S [16] | 23.8 | 5.2 | - | 81.3 |
| CvT-13 [46] | 20 | 4.5 | - | 81.6 |
| PVT-Small [45] | 24.5 | 3.8 | 820 | 79.8 |
| CPVT-Small-GAP [7] | 23 | 4.6 | 817 | 81.5 |
| Swin-T [27] | 29 | 4.5 | 766 | 81.3 |
| DS-Net-S (**ours**) | 19.7 | 3 | 582 | **81.9** |
| DS-Net-S$^*$ (**ours**) | 23 | 3.5 | 510 | **82.3** (+2.4) |
| DeiT-B [42] | 86 | 17.5 | 292 | 81.8 |
| CrossViT-18 [4] | 43.3 | 9 | 430 | 82.5 |
| ConTNet-B [47] | 39.6 | 6.4 | - | 81.8 |
| PVT-L [45] | 61.4 | 9.8 | - | 81.7 |
| Swin-S [27] | 50 | 8.7 | 437 | 83.0 |
| DS-Net-B (**ours**) | 48.8 | 7.6 | 387 | **82.8** |
| DS-Net-B$^*$ (**ours**) | 49.3 | 8.4 | 335 | **83.1** (+1.3) |

**Results.** As shown in Table 3, any absence of the components in DS-Block deteriorates the performance. Compared to extracting local and global features simultaneously, the model's representation capacity is constrained with either $f_L$ or $f_G$. Surprisingly, the unilateral co-attention of $w/o\ G \to L$ and $w/o\ G \to L$ performs worse than removing any fusion strategy, because implementing $G \to L$ would confuse the original local features $f_L$, and only by $L \to G$ could the missing local features be complemented. This further demonstrates the importance of the designed Inter-scale Alignment module.

Table 5: Comparison with the performance for object detection and instance segmentation on the MSCOCO minival set of other state-of-art methods.

| Backbone | Params(M) | $AP^b$ | $AP^b_S$ | $AP^b_M$ | $AP^b_L$ | $AP^s$ | $AP^s_S$ | $AP^s_M$ | $AP^s_L$ |
|---|---|---|---|---|---|---|---|---|---|
| RetinaNet [24] | | | | | | | | | |
| ResNet-50 [18] | 37.7 | 36.3 | 19.3 | 40.0 | 48.8 | - | - | - | - |
| Swin-T [27] | 38.5 | 41.5 | 25.1 | 44.9 | 55.5 | - | - | - | - |
| DS-Net-S* (**ours**) | 33.2 | **42.7** | **26.8** | **46.3** | **56.7** | - | - | - | - |
| Mask R-CNN [17] | | | | | | | | | |
| ResNet-50 [18] | 44.2 | 38.2 | 21.9 | 40.9 | 49.5 | 34.7 | 18.3 | 37.4 | 47.2 |
| Swin-T [27] | 48 | 43.7 | **28.5** | 47.0 | 57.3 | 39.8 | **24.2** | 43.1 | 54.6 |
| DS-Net-S* (**ours**) | 43.2 | **44.3** | 28.3 | **47.7** | **58.8** | **40.2** | 24.0 | **43.4** | **54.7** |

## 4.2 Image Classification

**Settings.** Image classification experiments are performed on ImageNet-1K [8], comprising 1.28M training images and 50K validation images of 1000 classes. For fair comparison with other works, we follow the training settings in DeiT. We train our model for 300 epochs by AdamW optimizer. The initial learning rate is set to 1e-3 and scheduled by the cosine strategy.

**Results.** We report the results in Table 4, where DS-Net and DS-Net* denote our proposed networks with and without Inter-scale Alignment module respectively. For DS-Net, only concatenation operation is used to fuse the local and global features. DS-Net* is the version with Inter-scale Alignment module. With comparable parameters and complexity, our proposed DS-Net consistently outperforms the DeiT [42] baseline by significant margins, which is, specifically, 6.8% improvement for tiny model, 2.4% improvement for small model and 1.3% for big model with only half parameters. The efficacy of Inter-scale Alignment module is also clearly indicated in Table 4, which gains further 0.9% , 0.4% and 0.3% improvement compared to the simple version of DS-Net-T, DS-Net-S and DS-Net-B. Notably, without bells and whistles, DS-Net-T* DS-Net-S* also outperform other state-of-the-art methods, including CNN, vision transformer and some hybrid architectures.

## 4.3 Object Detection and instance segmentation

**Settings.** Experiments in this part are conducted on the challenging MSCOCO 2017 [25], containing 118K training images and 5K validation images. Specifically, the backbone is initialized by pretraind weights from ImageNet-1K, and other layers adopt Xavier. We evaluate our DS-Net on typical detectors: RetinaNet and Mask R-CNN [17]. As the standard 1× schedule(12 epochs), we adopt AdamW optimizer with initial learning rate of 1e-4, decayed by 0.1 at epoch 8 and 11. We set stochastic drop path regularization of 0.1 and weight decay of 0.05. Following previous methods, training and testing images are resized to 800×1333.

**Results.** The results are shown in Table 5. For object detection, DS-Net-S* achieves significant improvement compared to ResNet-50 backbone [24] by 6.4% with RetinaNet and 6.1% with Mask R-CNN in terms of $AP^{bbox}$, and outperforms the previous state-of-the-art Swin-T [27] by 1.2% and 0.6%. For instance segmentation, DS-Net-S* achieves 40.2% in terms of $AP^{segm}$, which surpasses the ResNet-50 and Swin-T by 5.5% and 0.4%.

## 4.4 Dual-stream Feature Pyramid Networks

**Settings.** Experiments of DS-FPN are implemented on DS-Net-T and Swin Transformer [27] for object detection and instance segmentation on MSCOCO 2017 [25], by replacing the original FPN with DS-FPN. The training setting and scheme are the same as those mentioned above, which is 1× schedule(12 epochs) and AdamW optimizer with initial learning rate of 1e-4. Likewise, the backbone is initialized with pretrained weights, and DS-FPN is trained from scratch with newly added heads.

**Ablation Study.** In Table 6, we report results of several different designs combining DS-Blocks with FPN on RetinaNet, with DS-Net-T backbone. Four inserted positions of DS-Blocks have been

Table 6: Performance of DS-Blocks' different inserted positions on FPN of RetinaNet.

| Inserted Position | None | Last | Lateral | Lateral$_{rev}$ | Lateral$_{extra}$ |
|---|---|---|---|---|---|
| $AP^{bbox}$ | 39.0 | 40.2 | 40.3 | 40.2 | **40.4** |
| Params (M) | 18.83 | 19.59 | 23.72 | 21.36 | 24.43 |
| Flops (G) | 184.02 | 194.27 | 197.22 | 195.27 | 199.92 |
| Throughput (imgs/s) | 19.4 | 16.2 | 15.1 | 15.9 | 14.6 |

Table 7: Comparison on the performance of DS-FPN and FPN for object detection and instance segmentation on MSCOCO 2017.

| Backbone | Neck | $AP^b$ | $AP^b_S$ | $AP^b_M$ | $AP^b_L$ | $AP^s$ | $AP^s_S$ | $AP^s_M$ | $AP^s_L$ |
|---|---|---|---|---|---|---|---|---|---|
| | | | | RetinaNet [24] | | | | | |
| DS-Net-T | FPN | 39.0 | 23.3 | 42.6 | 51.3 | - | - | - | - |
| | DS-FPN | **40.4** | **26.1** | **44.2** | **52.3** | - | - | - | - |
| Swin-T [27] | FPN | 41.5 | 26.4 | 45.1 | **55.7** | - | - | - | - |
| | DS-FPN | **42.3** | **27.7** | **45.9** | 55.0 | - | - | - | - |
| | | | | Mask R-CNN [17] | | | | | |
| DS-Net-T | FPN | 40.1 | 24.9 | 43.4 | 52.1 | 37.2 | 21.4 | 40.4 | 50.0 |
| | DS-FPN | **41.3** | **25.8** | **44.6** | **53.0** | **37.9** | **22.1** | **41.1** | **50.4** |
| Swin-T [27] | FPN | 42.5 | 26.3 | 45.9 | 56.1 | 39.2 | 22.8 | 42.4 | 53.8 |
| | DS-FPN | **44.2** | **27.6** | **47.6** | **58.0** | **40.6** | **23.9** | **43.9** | **55.2** |

experimented, where "Last" means replacing the last $3\times3$ convolution layer, and "Lateral" equips them on lateral connections. More specifically, "Lateral$_{rev}$" moves the inner channel-alignment $1\times1$ convolution layer from the last place to the first, and "Lateral$_{extra}$" adds DS-Blocks also on extra feature outputs. The results show that "Lateral$_{extra}$" has the best performance. Therefore, we set "Lateral$_{extra}$" as our default, which is 1.4% higher than FPN with marginal memory costs.

**Comparison with FPN.** As reported in Table 7. For object detection, DS-Net-T and Swin Transformer with DS-FPN gain 1.4% and 0.8% mAP improvement, respectively. For instance segmentation, DS-FPN helps to gain 0.7% and 1.4% $AP^{segm}$ improvement for DS-Net-T and Swin Transformer, respectively. The results significantly prove the generality and effectiveness our Dual-stream design.

## 5    Conclusion

In this paper, we propose a generic Dual-stream Network, called DS-Net, which disentangles local and global features by generating two representations with different resolutions. We present Intra-scale Propagation module and Inter-scale Alignment module to combine the merits of convolution and self-attention mechanisms, and identify cross-scale relations between local and global tokens. Besides, for downstream tasks, we design a Dual-stream Feature Pyramid Network (DS-FPN) to introduce contextual information to feature pyramid for further refinement with marginal costs. With the outstanding performance on image classification and dense downstream tasks including object detection and instance segmentation, our proposed DS-Net has shown its promising potential in vision tasks.

## Disclosure of Funding

This work was supported by the National Natural Science Foundation of China under Grant 62076016 and the Shanghai Committee of Science and Technology, China (Grant No. 21DZ1100100).

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
