# OpenReview forum: "Dual-stream Network for Visual Recognition"
_NeurIPS.cc/2021/Conference — NeurIPS 2021 Poster_

### Official Review · Reviewer_tvGe · 2021-07-09

**Rating:** 5
**Confidence:** 4

**Summary:**

This paper proposes Dual-stream Network (DS-Net) for visual recognition. The two streams correspond to high-resolution convolutions for local features and low-resolution vision transformers for global patterns. The paper proposes an intra-scale propagation module to process the two streams and an inter-scale alignment module to fuse them into one. For object detection, it designs a Dual-stream FPN (DS-FPN) that adds dual-stream blocks to every feature pyramid. The proposed DS-Net achieves good results for image classification, object detection, and instance segmentation.

**Limitations And Societal Impact:**

The authors have adequately addressed them in the submission.

**Main Review:**

The paper is overall nicely organized. I have some comments that might need the authors’ attention.

1. The writing quality needs improvements. There are many grammar errors here and there. I would recommend the authors pay more attention to proofreading. The frequency of the writing errors is high and it concerns me. This factor is not considered when giving the ratings, though, because I believe they can be fixed.

2. The paper likes to claim that a statement is obvious. For example, in Line 38, the paper states that the previous works that attempt to introduce convolutions to vision transformers are obviously not the most ideal design. I personally don’t think it’s appropriate to claim they are obvious especially for topics that do not have a consensus. The other example is Line 172, stating that the naive fusion is not good enough obviously. However, according to the experimental results in Table 3 that compare DS-Net-S (naive) and DS-Net-S* (inter-scale alignment module), the proposed method outperforms the naive solution by 0.4% (81.9% → 82.3%) but at the cost of 16.7% additional computations (3.0 GFLOPs → 3.5 GFLOPs). Therefore, I wouldn’t think the naive fusion is not good enough obviously. By the way, there is an error when introducing DS-Net and DS-Net* in Line 251 -- “with” and “without” should be swapped.

3. The novelty is not strong. The idea of the dual-stream network is just to have one convolutional stream and one transformer stream. This is similar to dual-path network [a], which is missing in the related work. The idea of using different resolutions has also been explored, e.g. [b]. I wouldn’t think the novelty of this paper is particularly strong. In addition, after looking at Table 3, the improvements over the previous methods are also marginal.

In summary, I find the novelty and the performance are not that strong. Unfortunately, I wouldn’t be able to recommend acceptance for this paper. The authors should pay attention to the writing quality as well when revising this submission.

[a] Chen, Y., Li, J., Xiao, H., Jin, X., Yan, S. and Feng, J., 2017. Dual path networks. In NeurIPS 2017.

[b] Wang, H., Kembhavi, A., Farhadi, A., Yuille, A.L. and Rastegari, M., 2019. Elastic: Improving cnns with dynamic scaling policies. In CVPR 2019

**EDIT: Post-rebuttal**

Thanks for preparing a detailed response to the initial comments. After reading the response, the concerns about the novelty and performance improvements still remain. Moreover, as mentioned in the authors' response, when the model size is larger, the improvement over other baselines also becomes smaller -- 83.1% using 46M for the proposed method v.s. 83.0% with 50M of Swin-S. The proposed method also has less throughput in general. Unfortunately, I will keep the previous rating.

**Time Spent Reviewing:**

6

---

> ### Author Response · Authors · 2021-08-09
> **Replyr to reviewer tvGe**
>
> We would like to thank the reviewer for valuable comments and helpful suggestions.
>
> >Q1. The writing quality needs improvements. There are many grammar errors here and there. I would recommend the authors pay more attention to proofreading. The frequency of the writing errors is high and it concerns me. This factor is not considered when giving the ratings, though, because I believe they can be fixed.
>
> A1. Thank you, we will carefully proofread the paper and correct errors in the updated version.
>
> >Q2. The paper likes to claim that a statement is obvious. For example, in Line 38, the paper states that the previous works that attempt to introduce convolutions to vision transformers are obviously not the most ideal design. I personally don’t think it’s appropriate to claim they are obvious especially for topics that do not have a consensus. The other example is Line 172, stating that the naive fusion is not good enough obviously. However, according to the experimental results in Table 3 that compare DS-Net-S (naive) and DS-Net-S* (inter-scale alignment module), the proposed method outperforms the naive solution by 0.4% (81.9% → 82.3%) but at the cost of 16.7% additional computations (3.0 GFLOPs → 3.5 GFLOPs). Therefore, I wouldn’t think the naive fusion is not good enough obviously. By the way, there is an error when introducing DS-Net and DS-Net* in Line 251 -- “with” and “without” should be swapped.
>
> A2. Thank you, we have tempered our language in the updated version. The proposed Inter-scale Alignment Module indeed introduces additional computations. However, by visualizing weight maps $W_{G \rightarrow L}$ and $W_{L \rightarrow G}$ (defined in Section 3.3 in the paper). We find that to global features, the importance of different pixels of local features varies dramatically, and vice versa, which demonstrates that simply concatenating or adding them position-wisely is not the best solution. Inter-scale Alignment module helps achieve about 1\% and 0.4\% performance gain on DS-Net-T and DS-Net-S respectively (Table 3 in the paper). We will explore how to reduce the extra computation cost caused by such a module in the future work. We wil correct the error in Line 251 in the updated version.
>
> >Q3. The novelty is not strong. The idea of the dual-stream network is just to have one convolutional stream and one transformer stream. This is similar to dual-path network [a], which is missing in the related work. The idea of using different resolutions has also been explored, e.g. [b]. I wouldn’t think the novelty of this paper is particularly strong. In addition, after looking at Table 3, the improvements over the previous methods are also marginal.
>
> A3. Dual-path structure in [a] is proposed to reveal the equivalence of of ResNet [1] and DenseNet [2]. In this paper, different from previous works, dual-stream structure is firstly proposed to disentangle local and global representations, which helps maximize  their merits. In [b], using different resolutions is to help networks adapt to objects with various scales in training data. While in this paper, dual resolution is proposed to be compatible with different characteristics of convolution and self-attention. High resolution retains fine-grained local details, which is used for convolution; low resolution retains global patterns, which prevent self-attention mechanism from being overwhelmed by local details. Additionally, we first propose Inter-scale Alignment (Section 3.3 in the paper) to solve the misalignment issue between local and global features, which achieves 0.98\% and 0.4\% performance gains on DS-Net-T and DS-Net-S (Table 3 in the paper). Thus, this module is also a main contribution of this paper.  Besides, we  design DS-FPN for downstream tasks like detection and instance segmentation, which also achieves obvious performance gain.
>
> [1] K. He, X. Zhang, S. Ren, and J. Sun. Deep residual learning for image recognition. In IEEE CVPR, pages 770–778, 2016.
> [2] G. Huang, Z. Liu, L. Maaten, K. Weinberger. Densely Connected Convolutional Networks. In IEEE CVPR, pages 4700-4708, 2017.
>
> Best,
> Authors

---

### Official Review · Reviewer_cxxV · 2021-07-15

**Rating:** 8
**Confidence:** 4

**Summary:**

The authors propose a new architecture for visual recognition. Like standard convolutional architectures, the representation is structured in a multi-scale hierarchy, with deeper layers being coarser and more abstract. At every level of the hierarchy, a coarse, global representation is appended and simultaneously processed with the more fine-grained, local one. Local features are processed with depth-wise convolutions, and global ones with multi-head self-attention. In addition to this separate processing of local and global features, they are co-processed by cross-attending between local and global features.

In addition to this recognition backbone, the authors adapt Feature Pyramid Networks in a similar spirit. Whereas FPN's use dense convolutions to fuse an intermediate representation with a coarser one, the authors instead use the proposed "Dual-Stream" block, combining local depth-wise convolutions, global self-attention, and local/global cross-attention.

The authors evaluate the recognition backbone for classification on ImageNet and detection and instance segmentation on COCO, reporting large gains over standard convolutional architectures such as ResNet, and reasonable gains over more recent transformer-based ones.

**Limitations And Societal Impact:**

The authors propose a new architecture for visual recognition and evaluate it on standard benchmarks. As such the work does not raise any new ethical concerns.

**Main Review:**

The authors propose a new hybrid architecture which advances the state of the art in visual recognition. Rather than seeking to entirely replace the convolutional architecture with attentional ones, as prior work has, the authors introduce a careful combination of convolutional and attention-based operations. In particular, the main obstacle for introducing attention in visual architectures is the quadratic scaling of attention operations with respect to the number of tokens. The authors overcome this limitation by introducing low-resolution latents at every stage, and attending within or across these latents. Nevertheless, the architecture is simple and general enough to be used as a drop-in replacement to other visual backbones (such as ResNet), indicating the generality of the approach. The gains afforded by their FPN variant (which incorporates the same design principle) also support this generality. Together, these qualities make this a solid contribution.

**Time Spent Reviewing:**

2

---

> ### Author Response · Authors · 2021-08-09
> **Reply to reviewer cxxV**
>
> Thanks for the support of the novelty and results of our work. We will keep improving the draft.
>
> Best,
> Authors

---

### Official Review · Reviewer_LMod · 2021-07-17

**Rating:** 7
**Confidence:** 5

**Summary:**

This paper proposes a hybrid convolution+transformer model for 2d images. To summarize the model, it is broken up into 4 stages that operate at successively higher resolutions much like a typical Resnet.  At each stage, there are two parallel paths, one that apples a depthwise convolution at high resolution (capturing local effects), and one that applies a standard a multi-headed attention at coarse (capturing global effects) resolution.  These paths are fused by a final attention module which allows for local to attend to global and vice versa.

Another contribution is the DS-FPN (which simply slots one of the above “dual stream” modules into the lateral paths of a typical FPN). Experiments show strong performance on both image classification for ImageNet and detection/segmentation for COCO (relative to competing Transformer-only models like Deit).


**Limitations And Societal Impact:**

Yes.

**Main Review:**

Overall, the authors have proposed an intuitive and conceptually simple model that enjoys the benefits of both convolution and transformer/attention.  They achieve strong performance without an overly expensive model, which to my knowledge is novel. I have some complaints on exposition/writing but no major criticisms otherwise.  Based on good execution and reasonably strong results, I’m recommending acceptance.

Some more detailed comments/questions:
* In several places throughout the paper, the authors claim that something is “obvious” when it is not really obvious (and certainly not provable in a mathematical sense)… I recommend tempering this language.
* In the intro, the authors say “​​the conflicting properties of such two operations, convolution for local patterns but attention for global patterns, might cause ambiguity during training,...”: it’s not clear what this means and how one might quantify this statement.
* There is a sentence in the related work section: “progressively aggregate neighboring tokens before transformers via overlapping convolution-alike way to enhance their representations” that makes no sense.
* Related paper: “MaX-DeepLab: End-to-End Panoptic Segmentation with Mask Transformers” by Wang et al also use convolutions and multi-headed attention in a manner where both types of modules are woven fairly tightly together through the network. It’s quite a different approach in some ways, but should be cited here.
* Figure 2: is missing a 1x1 convolution at the end of the Inter-scale alignment block
* Is there a network stem that first does a reduction to stride 4 (as in Resnet)?
* I’m very surprised that only depthwise convolutions are used in the convolution path — and would like to clarification as to whether these are literally depthwise convolutions as stated in Eqn 1 or whether there is also a 1x1 convolution following (thus making it a depthwise separable convolution, which is far more commonly used in convnets).  Using a depthwise-only convolution is pretty limited, allows for no interaction between channels and essentially would amount to a blur operation, which is why this is so surprising.
    * Along the same lines, some papers have found it important to make the first convolution
* Clarification: are positional encodings used anywhere?  Would they help if not?
* For Table 3, I will just note that there are many more recent convolution-only networks than ResNet — I know that this is not necessarily the point, but it would be good to have a comparison against the most recent convolution-only networks to see how the transformer based models stand.
* For Table 4, I recommend adding in some other baselines, especially DETR (a transformer based detection model).
* In Section 4.3, authors claim to evaluate DS-Net in the Cascade R-CNN model, but these experiments do not seem to be in the paper.
* Why is DS-Net-T used in Table 6 instead of DS-Net-S?
* What is the running time impact of using DS modules in the FPN?


**Time Spent Reviewing:**

2

---

> ### Author Response · Authors · 2021-08-09
> **Repy to reviewer LMod**
>
> We thank the reviewer for the insightful comments and suggestions.
>
> >Q1. In several places throughout the paper, the authors claim that something is “obvious” when it is not really obvious (and certainly not provable in a mathematical sense)… I recommend tempering this language.
>
> A1. Following this suggestion, we will  temper our language in the updated version.
>
> >Q2. In the intro, the authors say “the conflicting properties of such two operations, convolution for local patterns but attention for global patterns, might cause ambiguity during training,...”: it’s not clear what this means and how one might quantify this statement.
>
> A2. By visualizing weight maps of self-attention mechanism, we find that the highly activated pixels are sporadic, which contains relations between different objects, while convolution operations mainly focus on local details. Compared to cascade structure, which may constrain the performance of both attention and convolution, dual-stream structure disentangles local and global representations, which helps maximize both of their merits.
>
> >Q3. There is a sentence in the related work section: “progressively aggregate neighboring tokens before transformers via overlapping convolution-alike way to enhance their representations” that makes no sense.
>
> A3. Thank you, we will change it to "progressively aggregate neighboring tokens to one token to capture local structure information, similar to convolution operation" in the updated version.
>
> >Q4. Related paper: “MaX-DeepLab: End-to-End Panoptic Segmentation with Mask Transformers” by Wang et al also use convolutions and multi-headed attention in a manner where both types of modules are woven fairly tightly together through the network. It’s quite a different approach in some ways, but should be cited here.
>
> A4. Thank you, we will cite this paper in the updated version
>
> >Q5. Figure 2: is missing a 1x1 convolution at the end of the Inter-scale alignment block.
>
> A5. Thank you, we will correct it in the updated version.
>
> >Q6. Is there a network stem that first does a reduction to stride 4 (as in Resnet)?
>
> A6. No. Similar to ViT [1], we only implement down-sampling in patch embedding modules.
>
> >Q7. I’m very surprised that only depthwise convolutions are used in the convolution path — and would like to clarification as to whether these are literally depthwise convolutions as stated in Eqn 1 or whether there is also a 1x1 convolution following (thus making it a depthwise separable convolution, which is far more commonly used in convnets). Using a depthwise-only convolution is pretty limited, allows for no interaction between channels and essentially would amount to a blur operation, which is why this is so surprising.
>
> A7. Indeed, there is no 1x1 convolution following depth-wise convolutions. We only implement 1x1 convolution before Dual-scale representations (Section 3.1 in the paper) and after Inter-scale alignment block (Section 3.3 in the paper).
>
> >Q8. Clarification: are positional encodings used anywhere? Would they help if not?
>
> A8. We use positional encodings at the beginning of each blocks. The classification performance on ImageNet-1k is almost the same without positional encodings (about 0.1\% drop). It is probably because convolution operation itself contains positional information.
>
> >Q9. For Table 3, I will just note that there are many more recent convolution-only networks than ResNet — I know that this is not necessarily the point, but it would be good to have a comparison against the most recent convolution-only networks to see how the transformer based models stand.
>
> A9. Thank you, we will add more convolution-only networks in the updated version.
>
> >Q10. For Table 4, I recommend adding in some other baselines, especially DETR (a transformer based detection model).
>
> A10. Following the suggestion, we are conducting corresponding experiments and the results will be added in the updated version.
>
> >Q11. In Section 4.3, authors claim to evaluate DS-Net in the Cascade R-CNN model, but these experiments do not seem to be in the paper.
>
> A11. Sorry for  confusion, we did not evaluate DS-Net in the Cascade R-CNN model. We will correct this writing error in the updated version.
>
> >Q12. Why is DS-Net-T used in Table 6 instead of DS-Net-S?
>
> A12. We choose DS-Net-T to save experimental time and thus obtain more results. We also test DS-FPN on RetinaNet with DS-Net-S and achieves 0.8\% improvement in terms of mAP compared to FPN.
>
> >Q13. What is the running time impact of using DS modules in the FPN?
>
> A13. We test the latency as suggested in the following table.  The ${Last}$ design achieves 1.2\% improvement in terms of mAP with negligible extra parameters (0.76M) and latency (19->16img/s), a good trade-off between performance and costs. Note that we provide several different designs of DS-Blocks as illustrated in Table 5 in the paper.
>
> ##
> **Comparison of mAP, Latency, Params and Flops of different DS-FPN designs.**
>
> | Inserted Position |(None) |  (Last) | (Lateral$_{extra}$)|
> | :---------: | :------------: | :----------------: | :--------------: |
>  |$AP^{bbox}$  |39  |40.2  |40.4  |
>  |Params (M)   |18.83  |19.59  |24.43  |
>  |Flops (G)   |184.02  |194.27  |199.92  |
>  |Throughput (img/s)   |19.4  |16.2  |14.6  |
>
> ##
> [1] A. Dosovitskiy, L. Beyer, A. Kolesnikov, D. Weissenborn, X. Zhai, and T. U. et al. An image is worth 16x16 words: Transformers for image recognition at scale. In ICLR, 2021
>
> Best,
> Authors

---

### Official Review · Reviewer_ybP5 · 2021-07-21

**Rating:** 5
**Confidence:** 5

**Summary:**

This paper presented a Dual-stream Network for image classification. It combines the representation of local and global pattern features by using self-attention and convolution together. It also proposed an Intrascale Propagation module to process two different resolutions in each block.  In addition, it also introduced such block to FPN to demonstrate the benefit.

**Limitations And Societal Impact:**

Please justify the weakness to improve the paper.



**Main Review:**

This paper is easy to read. The methodology and its novelty are clear. However, the setup of experiments are problematic and confusing.

Strength:
+ The proposed Intra-scale Propagation module is novel. Combing SA with Convolution makes sense.
+ The performance and speed is comparable to concurrent works.

Weakness: There are some issues needed to be addressed before accepted.
- In Table 3, the model setup for tiny and small model is confusing. It is not aligned with competitors on parameters and flops, which make hard to do fairly comparison.
- It lacks the performance comparison on large models. Will the performance hold?
- It lacks necessary ablation study on the proposed Intra-scale Propagation module. Based on Table 3, the improvement of such module is small. It is also decreasing when model size increases. The author needs to justify the effectiveness of this module.
- The benefit of DS-FPN is not convincing. Although it shows improvement in Table 4, it also significantly increases the number of parameters. How about the latency?
- Writing issues: The precision used in reporting numbers is not consistent.  The presentation of tables (e.g. font, whitespace) needs improvement.

Post-rebuttal:
Thanks authors for providing more details. However,  the newly provided ablation study doesn’t justify the effectiveness of proposed components. Hence, I will hold my rating as is.


**Time Spent Reviewing:**

2

---

> ### Author Response · Authors · 2021-08-09
> **Reply to Reviewer ybP5**
>
> We thank the reviewer for helpful comments and suggestions very much.
>
>
> >Q1.  In Table 3, the model setup for tiny and small model is confusing. It is not aligned with competitors on parameters and flops, which make hard to do fairly comparison.
>
> A1. The size of the small model is defined according to the commonly used standard (about 25M params), such as PVT-Small [1] (24.5M params) and TNT-Small [2] (23.8M params). As for tiny models, there is no consensus yet. For instance, the mode size of Swin Transformer-Tiny [3] and Deit-Tiny [4] are respectively  29M and  6M. DS-Net-Tiny (10.5M params) is given in terms of  ResNet-18 (11.8M params)[5] for a fair comparison.
>
> >Q2. It lacks the performance comparison on large models. Will the performance hold?
>
> A2. We have  tested the classification performance of large models (46M parms), which achieves 83.1\% top-1 accuracy on ImageNet-1k. The detection and segmentation performance will be also tested soon.
>
> >Q3. It lacks necessary ablation study on the proposed Intra-scale Propagation module. Based on Table 3, the improvement of such module is small. It is also decreasing when model size increases. The author needs to justify the effectiveness of this module.
>
> A3. We conduct corresponding ablation study to testify the effectiveness of different components in DS-Block, including two pathway of Intra-scale Propagation and bidirectional co-attention of Inter-scale Alignment. We respectively remove one of the four modules in every block of DS-Net-T$^\ast$, while remain the others unchanged. The results are shown in  Table 1, where $w/o$ $f_{L}$ denotes that $f_{l}$ (defined in Section 3.2 in the paper) is directly fed to Inter-scale Alignment Module without depth-wise convolutions, and $w/o$ $G\rightarrow L$ denotes that co-attention from global features to local features is not implemented, analogous to $w/o$ $f_{G}$ and $w/o$ $L\rightarrow G$. The version $w/o$ $L\leftrightarrow G$ removes the entire co-attention, which equals DS-Net-T.
> ##
> **Ablations of removing components of DS-Net-T$^\ast$ on ImageNet-1k validation set.**
>
> | Versions  | DS-Net-T$^\ast$ |  ($w/o$ $f_{L}$ ) | ($-w/o$ $f_{G}$) | ($w/o$ $G\rightarrow L$) |    ($w/o$ $L\rightarrow G$) | ($w/o$ $L\leftrightarrow G$)|
> | :---------: | :-----------------------------: | :------------: | :--------------: | :-----------------: | :-----------------: | :-----------------: |
> |   Top-1(\%)  | **79.04**  | 76.65  | 76.64  | 76.53 | 76.38 | 78.06|
> |Top-5(\%)  |**94.75** |93.61 |93.67 |93.45 |93.40 |94.13|
>
> ##
>
>
> The results show that any absence of the components in DS-Block deteriorates the performance. Compared to extracting local and global features simultaneously, the model's representation capacity is constrained with either $f_{L}$ or $f_{G}$. Surprisingly, the unilateral co-attention of $w/o$ $G\rightarrow L$ and $w/o$ $G\rightarrow L$ performs worse than removing any fusion strategy, because implementing $G\rightarrow L$ would confuse the original local features $f_{L}$, and only by $L\rightarrow G$ could the missing local features be complemented. This further demonstrates the importance of the designed Inter-scale Alignment module.
>
> >Q4. The benefit of DS-FPN is not convincing. Although it shows improvement in Table 4, it also significantly increases the number of parameters. How about the latency?
>
> A4. We test the latency as suggested in the following table.  Note that we provide several different designs of DS-FPN as illustrated in Table 5 in the paper.  The ${Last}$ design achieves 1.2\% improvement in terms of mAP with negligible extra parameters (0.76M) and latency (19->16img/s), a good trade-off between performance and costs.
>
> ##
> **Comparison of mAP, Latency, Params and Flops of different DS-FPN designs.**
>
> | Inserted Position |(None) |  (Last) | (Lateral$_{extra}$)|
> | :---------: | :------------: | :----------------: | :--------------: |
>  |$AP^{bbox}$  |39  |40.2  |40.4  |
>  |Params (M)   |18.83  |19.59  |24.43  |
>  |Flops (G)   |184.02  |194.27  |199.92  |
>  |Thoughput (img/s)   |19.4  |16.2  |14.6  |
>
> ##
>
> >Q5. Writing issues: The precision used in reporting numbers is not consistent. The presentation of tables (e.g. font, whitespace) needs improvement.
>
> A5. Thank you, these typos will be corrected in the final version.
>
> [1] W. Wang, E. Xie, X. Li, D.-P. Fan, K. Song, and D. L. et al. Pyramid vision transformer: A versatile backbone for dense prediction without convolutions. arXiv:2102.12122, 2021.
> [2] K. Han, A. Xiao, E. Wu, J. Guo, C. Xu, and Y. Wang. Transformer in transformer. arXiv:2103.00112, 2021.
> [3] Z. Liu, Y. 352 Lin, Y. Cao, H. Hu, Y. Wei, and Z. Z. et al. Swin transformer: Hierarchical vision transformer using shifted windows. arXiv:2103.14030, 2021.
> [4] H. Touvron,M. Cord,M. Douze, F.Massa, A. Sablayrolles, and H. Jégou. Training data-efficient image transformers distillation through attention. arXiv:2012.12877, 2020.
> [5] K. He, X. Zhang, S. Ren, and J. Sun. Deep residual learning for image recognition. In IEEE CVPR, pages 770–778, 2016.
>
> Best,
> Authors

---

### Decision · Program_Chairs · 2021-09-27

**Decision:**

Accept (Poster)

**Comment:**

The reviews are split between accept and borderline reject recommendations. Two reviewers with positive ratings (7 and 8) appreciated that the hybrid CNN+Transformer approach is simple and intuitive. They also liked that the paper reports good empirical performance. The other two reviewers with negative ratings (5 and 5) raised concerns about limited novelty and insufficient/unconvincing empirical results. They criticized that the novelty isn't particularly strong because prior have already looked into the dual-stream approach combining different architectures as well as multi-resolution processing. They also shared a common concern that there is insufficient empirical evidence showing that the proposed approach is truly beneficial compared to simpler alternatives, especially at a large-scale regime, such as Swin Transformer (Swin-S) that achieves similar performance at a comparable parameter size (Swin-S: 83% w/ 50M params vs. this work: 83.1% w/ 46M params). Separate from novelty/experiment concerns, all four reviewers raised several writing issues.

This meta-reviewer carefully read the reviews, rebuttal, post-rebuttal discussion, and the paper in detail to fully understand the concerns raised by the reviewers. I agree with reviewers cxxV and LMod that the proposed idea of combining convolution and self-attention operations is interesting and novel --  it is a simple and effective, yet non-trivial combination than simply stacking CNN and Transformer side-by-side. Plus, as the reviewer cxxV pointed out, the paper reports impressive transfer results on ImageNet and COCO, the latter of which is particularly interesting because it is far from being solved (compared to ImageNet). I do however share the concerns with tvGe and ybP5, and including additional results as requested by the two reviewers will make the paper much stronger. The authors are urged to incorporate all the feedback provided by the reviews in their final version.